# Numerical Analysis for Heat Transfer Augmentation in a Circular Tube Heat Exchanger Using a Triangular Perforated Y-Shaped Insert

**Lokesh Pandey** [1] **and Satyendra Singh** [2,*]

1    Uttarakhand Technical University, Dehradun 248007, Uttarakhand, India; lokesh.pandey21@gmail.com
2    Department of Mechanical Engineering, B.T. Kumaon Institute of Technology,
     Dwarahat 263653, Uttarakhand, India
*    Correspondence: ssinghiitd@gmail.com

**Abstract:** The present investigation constitutes CFD analysis of the heat transmission phenomenon in a tube heat exchanger with a Y-shaped insert with triangular perforation. The analysis is accomplished by considering air as a working fluid with a Reynolds number ranging from 3000 to 21,000. The segment considered for analysis consists of a circular tube of 68 mm diameter and 1.5 m length. The geometrical parameter considered is the perforation index (0%, 10%, 20%, and 30%). The constant heat flux is provided at the tube wall and a pressure-based solver is used for the solution. The studies are performed for analyzing the effects of inserts on the heat transfer and friction factor in the circular tube heat exchanger which results in augmented heat transfer at a higher perforation index (PI) and lower friction factor. The investigation results show that the highest heat transfer is 5.84 times over a simple plain tube and the maximum thermal performance factor (TPF) is 3.25 at PI = 30%, Re = 3000.

**Keywords:** heat exchanger; thermal performance; geometrical parameters; perforation; CFD





## 1. Introduction

Nowadays, energy has become an integral part of life for human beings. Researchers are continuously carrying out research to improve the efficiency of devices. The heat exchanger (HE) is a device to exchange heat energy from one source to another and plays an important role in many industries. It has applications in industries such as power plants, refrigeration, air-conditioning, chillers, and many more. Hence, it requires improving the performance of HEs. The heat transfer rate is increased by increasing the surface area and roughness, and by significant change in the boundary conditions. The heat transmission process is executed in three ways: active, passive, and compound. In the active technique, there is a transfer of heat with the assistance of external equipment. In the passive method, the augmentation of heat transfer is completed with the help of the placement of the secondary heating surface—i.e., inserts/turbulators. The employment of both techniques together is carried out in the compound category.

In an experimental investigation, Akcayoglu [1] studied ducts with half delta wings of double row pairs with two different configurations. Double counter-twisted tapes used by Bhuiya et al. [2] in their investigation demonstrated improvements in thermal performance with the increase in the twist ratio of mild steel. Promvonge et al. [3] used inclined vortex rings for investigation and found a 1.4 thermal performance factor at their lowest Reynolds number. Kumar et al. [4] found a maximum 1.47-times increment in thermal performance over a plain tube in their investigation of perforation in a circular ring with different perforation indexes. Zong et al. [5] used a hollow cross-disk insert for their numerical studies of the HE; as a results, they found that the flow nature was three-dimensional with turbulent flow. Hameed et al. [6] studied rectangular boxes and obtained that, for maximum outlet temperature, four inserts are to be placed in the fluid domain. Nagarajan

and Sivashanmugam [7] used a right–left helical spacer insert fitted in a circular tube in which they varied the twist ratio of the helix. They achieved improvement in heat transfer by increasing the value of Re and decreasing the value of the twist ratio. The study of an airfoil shaped insert by Gururatana et al. [8] revealed that the maximum TPF found was 1.45 when inserts were placed in 450. Aliabadi et al. [9] investigated a set of arrangements of delta winglets. They arranged 14 vortex-generator inserts with a forward and longitudinal arrangement of the delta winglets. The results showed that two-side-cut delta wings provide better heat transmission enhancement. The literature survey concludes that increasing the effectiveness of circular tube heat exchanger core surface disturbances plays an important role.

Double-sided delta-wing tape inserts have been investigated by Eiamsa-ard and Promvonge [10], in which they concluded that delta wings enhance both the heat transmission and performance of HE. In other studies, Eiamsa-ard and Promvonge [11] analyzed the performance of HE using counter-clockwise and clockwise twisted tape alternatively and used short-length twisted tapes [12] which generated strong swirl flow at the entry and improved the performance. Similarly, considering twin delta-winged twisted tape [13], coupling twisted tape [14], rectangular-cut twisted tapes used by Nakhchi and Esfahani [15], and using jagged twisted tape inserts by Rahimi et al. [16] were studied and achieved a reduction in pressure loss and higher fluid mixing due to higher vortex generation and obtained improved heat transfer characteristics. Singh et al. [17] experimentally investigated solid-hollow circular disks with rectangular winglets and noticed a thermal performance improvement in heat transmission over a simple plain tube (smooth circular tube without any type of insert). Similarly, researchers using circular-ring turbulators [18] and circular rings with twisted tape inserts [19] found enhancement in the heat transfer rate due to the formation of eddies in the former and due to the formation of swirl in addition to eddies in the latter. Gautam et al. [20] investigated using a perforated triple wing vortex generator; they found enhancement in heat transmission and thermal performance over a simple plain tube HE.

It is observed from the literature study that the inserts or turbulators help to reduce the friction factor near the inner wall surface and enhance the heat transfer. A triangular perforated Y-shaped insert in the test section may be one such type of turbulator for the study. The insert divides the fluid flow in three chambers of the Y-shape that reduces the fluid flow in the central core region; each chamber delivers fluid independently and perforation may create swirl that helps to reduce the thermal resistance, ensuring proper mixing of the fluid and enhancing the heat transfer. The numerical analysis of the present model has been carried out in ANSYS Fluent and a turbulence k-$\varepsilon$ model has been used. In this computational research, the effect of the perforated Y-shaped insert on *Nu*, *f*, and *TPF* has been presented.

## 2. Geometry of Insert

Y-shaped inserts with triangular perforations have been used as turbulators. The perforations made are isosceles triangles and possess four different cases of perforation indexes viz. perforation index (PI) = 0%, 10%, 20%, and 30%. The physical model of the test segment consists of a circular channel with a 34-mm internal radius and 1500-mm length, as shown in Figure 1a,b. The Y shape is in three wings and aligned at 120 degrees to each other with a height of 30 mm throughout the length, which is perforated in a triangular shape in each of its wings of three cases of the perforation index (PI)—10%, 20%, and 30%. A non-perforated case has also been studied (PI = 0%). In the present analysis, the air is considered as a working fluid with the temperature as 300 K and a constant wall heat flux density (Q) = 1000 W/m$^2$ applied throughout the wall of the test section. The air properties are shown in Table 1. The analysis has been performed for varying the Reynolds number ranging from 3000 to 21,000 for different cases of triangular perforated Y-shaped inserts and the non-perforated case.

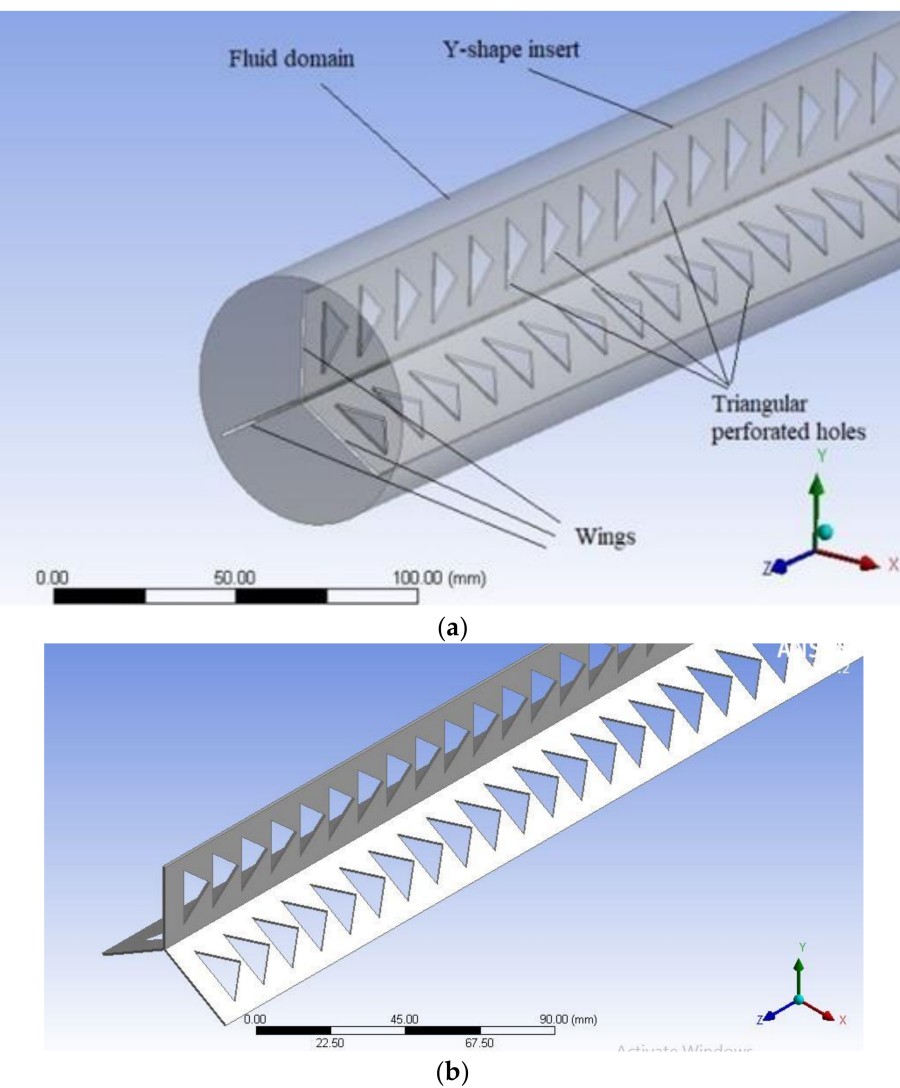

**Figure 1.** (**a**): Physical model of triangular perforated Y-shaped insert inside a circular tube; (**b**) triangular perforated Y-shaped insert for PI = 30%.

**Table 1.** Boundary condition and physical properties of air.

| Boundary Condition | Values |
| --- | --- |
| Air temperature at inlet (K) | 300 |
| Inlet velocity (m/sec) | 5.18, 4.44, 3.706, 2.963, 2.22, 1.48, 0.7409 |
| Heat flux (W/m$^2$) | 1000 |
| Ambient Pressure (atm) | 1 |
| Density (kg/m$^3$) | 1.225 |
| Specific heat (J/kg$^{-K}$) | 1006.53 |
| Viscosity (kg/m$^{-s}$) | $1.7894 \times 10^{-5}$ |
| Thermal conductivity (W/m$^{-K}$) | 0.0242 |

## 3. Mathematical Calculations

The expressions used in this investigation are given as:
Reynolds number:

$$Re = \rho(vD)/\mu \tag{1}$$

Nusselt number:

$$Nu = \frac{hD}{k} \tag{2}$$

Heat transfer coefficient:

$$h = \frac{Q}{(T_{wm} - T_{fm})A} \tag{3}$$

$$h = \frac{(T_o - T_i)mc_p}{(T_{wm} - T_{fm})A} \tag{4}$$

where

$$T_{wm} = (1/A) \int_0^A (T)dA \tag{5}$$

$$T_{fm} = (T_o - T_i)/2 \tag{6}$$

For calculating the friction factor [16]:

$$f = \frac{\Delta P}{(\frac{\rho v^2}{D}) * (\frac{L}{2})} \tag{7}$$

The effectiveness of HE is measured in the form of the thermal performance factor (*TPF*) [21]:

$$TPF = \left(\frac{Nu}{Nus}\right) / \left(\frac{f}{f_s}\right)^{\frac{1}{3}} \tag{8}$$

The perforation index (*PI*) determined as:

$$PI = \frac{Area\ of\ Perforated\ holes\ on\ surface}{Total\ area\ of\ the\ surface} \tag{9}$$

## 4. Modelling and Grid Independence Test

### 4.1. Mathematical Modelling

The incompressible fluid flow is assumed to be at constant fluid flow properties and heat due to the radiation being assumed to be zero. The equations used for the model are given as follows [15]:

Continuity Equation:

$$\frac{\partial(\rho u_i)}{\partial x_i} = 0 \tag{10}$$

Momentum Equation:

$$\frac{\partial(\rho u_i u_j)}{\partial x_j} = -\frac{\partial p}{\partial x_i} + \frac{\partial}{\partial x_i}(\mu \frac{\partial u_i}{\partial x_i} - \rho \overline{u_i' u_j'}) \tag{11}$$

Energy Equation:

$$\frac{\partial(\rho u_i T_{fm})}{\partial x_i} = \frac{\partial}{\partial x_i}(k\frac{\partial T_{fm}}{\partial x_i}) \tag{12}$$

### 4.2. Numerical Model

A numerical investigation has been carried out to understand the improvement of the heat transmission mechanism in circular HE having triangular perforated Y-shaped inserts using the "Ansys" module in which the k-ε turbulence model has been used for calculation with the finite volume method. For this investigation, a 1.5-m-long section has been used for the testing, in which 1000 W/m² constant heat flux was provided at the tube wall. A pressure-based solver was employed for solving the equations. High Reynolds numbers between 3000 and 21,000 have been taken for this analysis based on the RNG k-ε model.

The boundary conditions and physical properties of air considered in the investigation are shown in Table 1.

### 4.3. Grid Independence Test (GIT)

The grid independence test has been conducted for determining the number of elements presented in Table 2. The deviation for 15, 24, 877 elements in the Nusselt number was 0.2% and the friction factor was 4.59%, which is satisfactory in terms of accuracy for the simulation. Hence, further computational analysis was performed, taking 15, 24, 877 elements. The meshing of the insert with triangular perforation is shown in Figure 2.

**Table 2.** Grid independence test.

| Grid Number | Nu | $\left\lvert\frac{Nu_{i+1}-Nu_i}{Nu_{i+1}}\right\rvert$ | f | $\left\lvert\frac{f_{i+1}-f_i}{f_{i+1}}\right\rvert$ |
|---|---|---|---|---|
| 50, 258 | 35.87 | - | 0.1486 | - |
| 98, 547 | 48.27 | 0.2569 | 0.1857 | 0.2001 |
| 3, 54, 782 | 60.12 | 0.1971 | 0.2106 | 0.1181 |
| 8, 84, 704 | 62.48 | 0.0378 | 0.2302 | 0.0852 |
| 15, 24, 877 | 62.49 | 0.0002 | 0.2413 | 0.0459 |

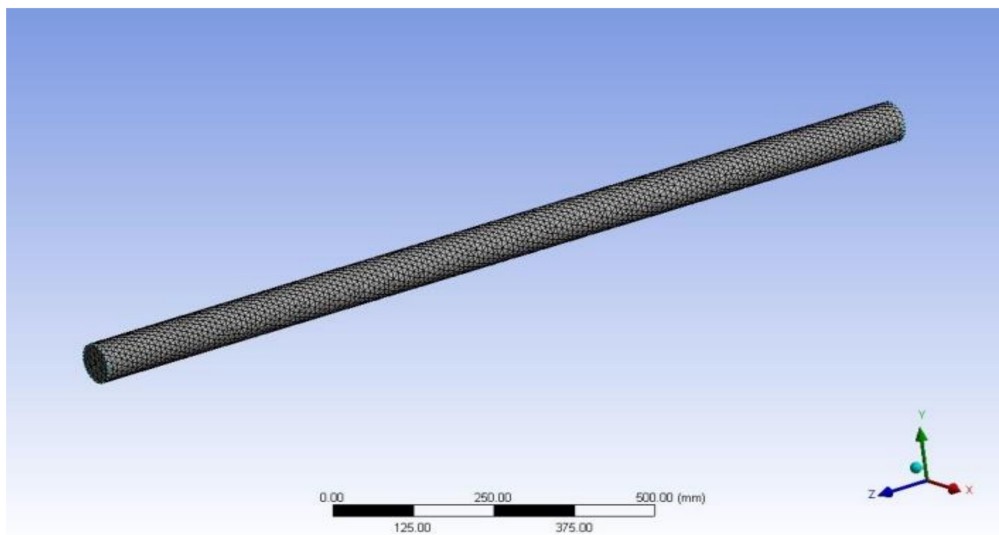

**Figure 2.** Meshing in Y-shaped insert with triangular perforation.

### 4.4. Model Validation

In order to verify the authenticity of the numerical model, the results for the smooth circular tube have been experimentally validated. The values obtained from numerical model of Nusselt number and friction factor are compared with the standard Dittus-Boelter equation and Blasius equation, respectively. The Dittus–Boelter equation and Blasius equation for the Nusselt number and friction factor are given in Equations (13) and (14), respectively.

$$Nu = 0.023 \times [\![\mathrm{Re}]\!]^{0.8} \times [\![\mathrm{Pr}]\!]^{0.4} \tag{13}$$

$$f = 0.316 \times [\![\mathrm{Re}]\!]^{(-0.25)} \tag{14}$$

The obtained results are presented in Figure 3. It can be seen in the figure that the obtained values of the Nusselt number and friction factor have shown good agreement with the experimental results with a deviation of ±4% and ±6%, respectively.

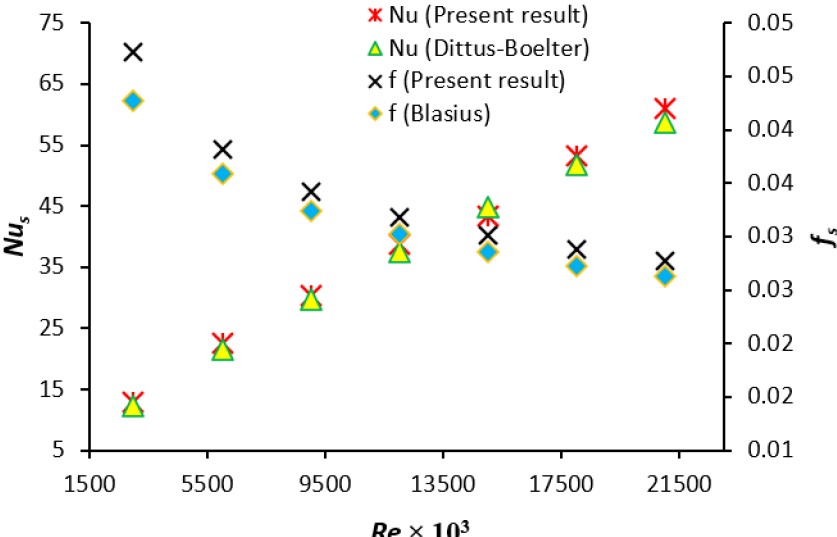

**Figure 3.** Validation of smooth tube result with standard equations for *Nu* and *f*.

## 5. Results

### 5.1. Effect on Nu

In Figure 4, variation of Nu with Re shows that as we increase the Reynolds number, the Nusselt number increased gradually. For the 30% perforation index case, the enhancement in the Nusselt number is 5.84 times over a simple plain tube. The increment in Nu is a result of increasing swirl for the higher range of Re, which enhanced fluid mixing inside the round tube. As Re increases, turbulence increases, which results in increment in the value of Nusselt number. The enhancement of Nu (rough tube) with respect to Nus (simple plain tube) is shown in Figure 5, which also reflects the similar nature of enhancement of heat transfer for the more perforated insert.

The temperature distribution created at the wall of tube due to Y-shaped triangular perforated insert and variation of temperature at the outlet are shown in Figure 6.

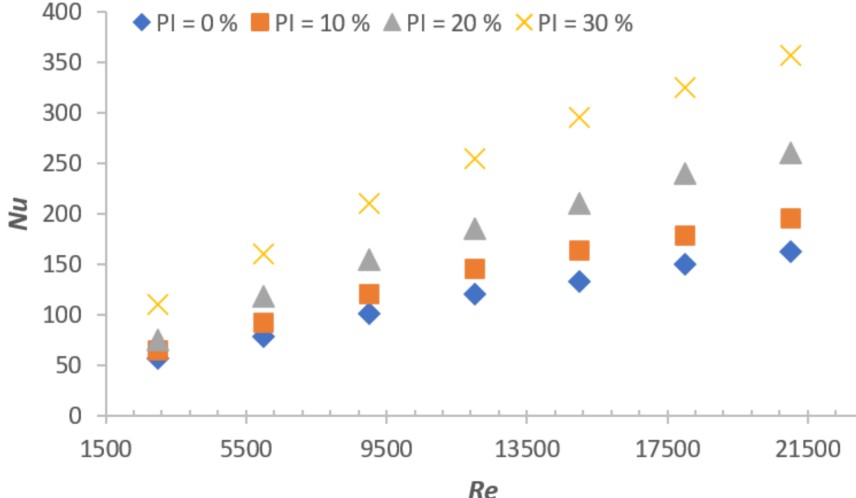

**Figure 4.** Variation of *Nu* with *Re*.

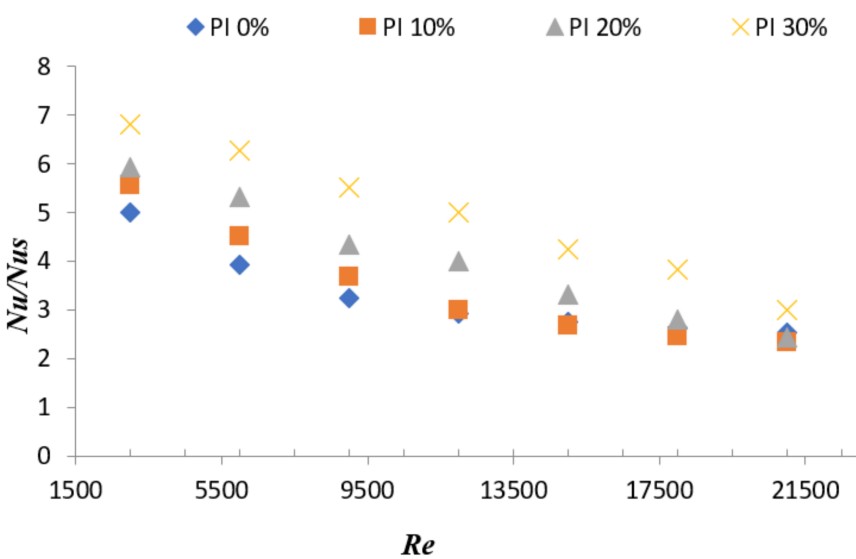

**Figure 5.** Variation of *Nu/Nus* with *Re*.

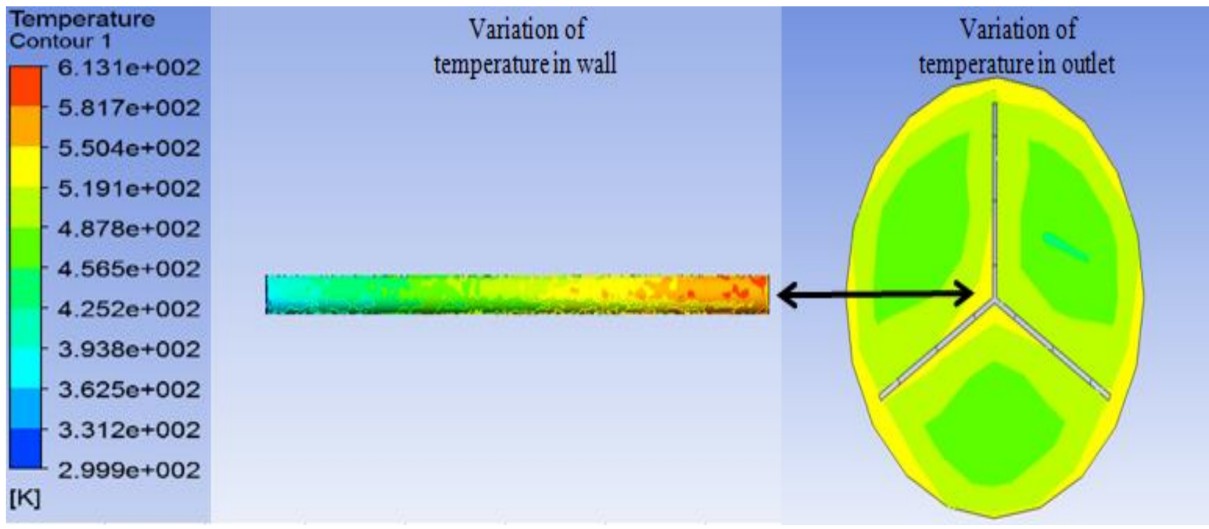

**Figure 6.** Temperature contours at wall and outlet.

### 5.2. Effect on f

The friction factor alters the thermal performance of HE. It is expected that the higher the friction factor, the lower the performance of HE. In case of PI = 20%, the friction factor was found higher at the Reynolds number between 3000 and 21,000. This is similar in case of the 10% perforation, the reason being the higher friction factor due to hole variation in the perforation of Y-shaped insert. Variation of *f* with Re is presented in Figure 7; however, comparison of *f* (rough tube) with respect to $f_s$ (smooth tube) is shown in Figure 8, in which a similar nature is shown for 20% perforation.

### 5.3. Effect on TPF

The maximum thermal performance (TPF) was found to be 3.25 in the case of the 30% perforation index at Reynolds number 3000. It can be noticed in Figure 9, that lower Re results in better heat transmission performance. This happens because at higher velocities, the test section undergoes a higher pressure drop; hence, the friction factor increases and thermal performance decreases.

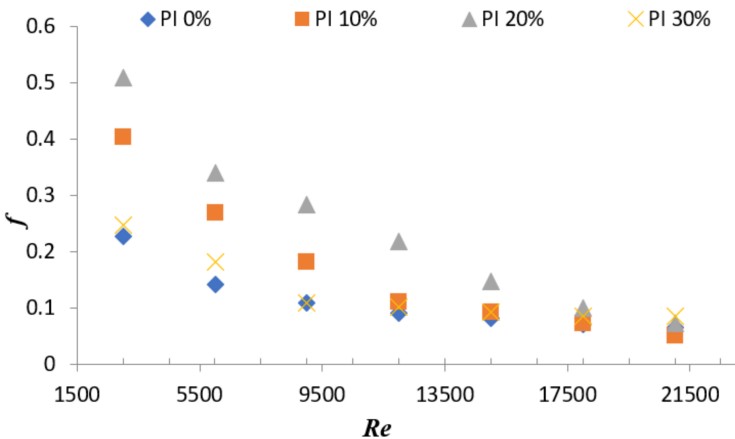

**Figure 7.** Variation of *f* with *Re*.

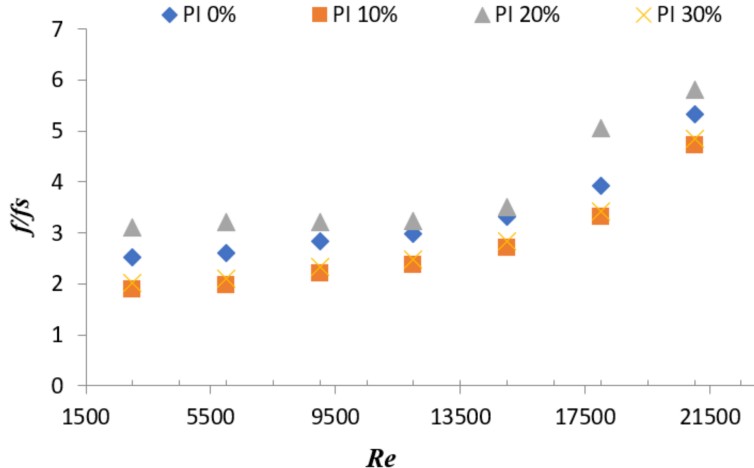

**Figure 8.** Variation of *f/fs* with *Re*.

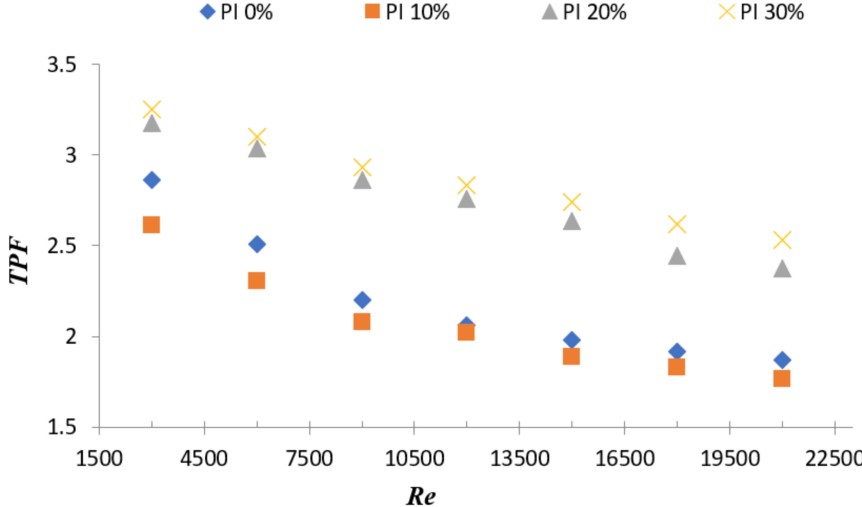

**Figure 9.** Variation of *TPF* with *Re*.

## 6. Comparison with Previous Work

In Figure 10, comparison of the present study geometry—a Y-shaped insert with triangular perforation—with previously published works has been carried out. The comparison is made in the same range of Reynolds number. The research works considered are: Bhuyia et al. [2], using double counter-twisted tape inserts; Promvonge et al. [3], using inclined vortex rings; Kumar et al. [4], using circular rings; Eiamsa-ard et al. [9], using a double-sided delta wing tape insert; Singh et al. [16], using a solid-hollow circular disk with rectangular winglets; and Gautam at al. [19], using a perforated triple-wing vortex generator. It can be seen from Figure 10, that the Y-shaped insert with triangular perforation provides the highest thermal performance factor among all others set of geometries of previous researchers.

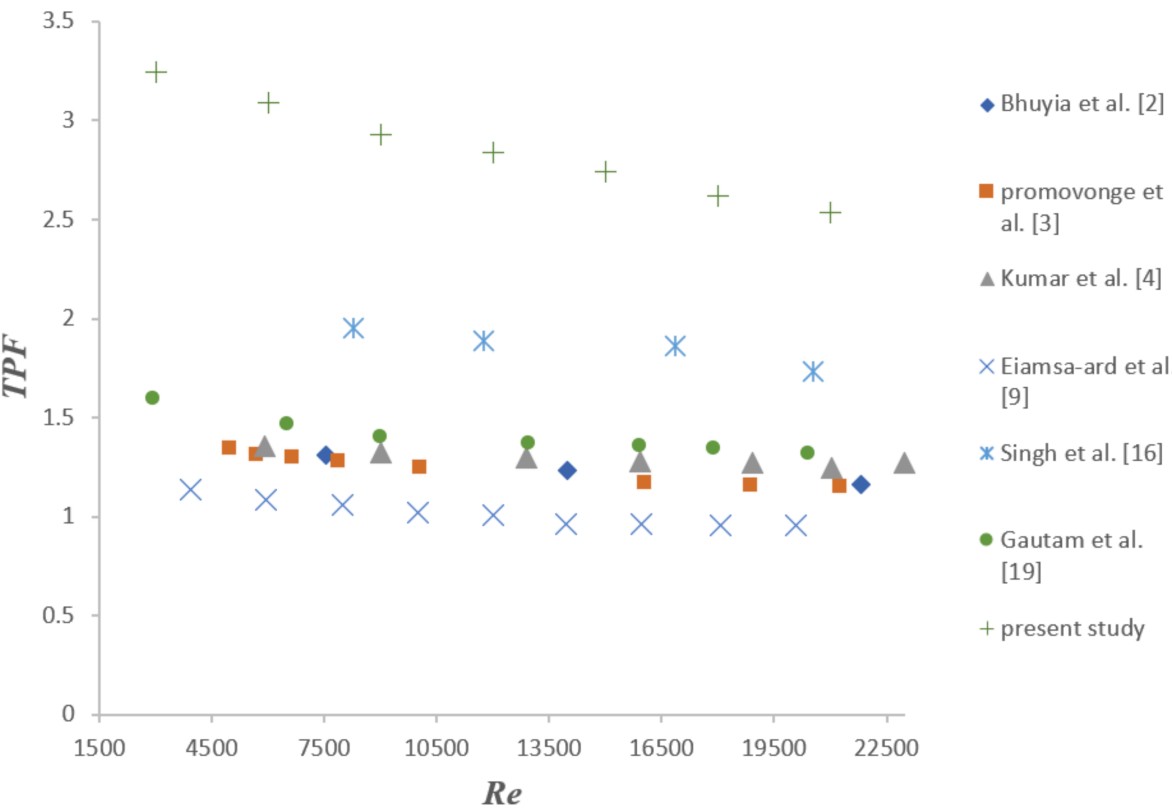

**Figure 10.** Comparison of triangular perforated Y-shaped insert with previously published works.

## 7. Conclusions

Under the present work, numerical investigation has been carried out considering the triangular perforated Y-shaped insert in the circular tube HE. Based on the obtained results, it can be said that there are a significant effects of the perforation ratio of triangular perforated Y-shaped inserts on $Nu$, $f$, and $TPF$. The major conclusions drawn from the study are as follows:

- The results show that increasing Re provides better heat transmission characteristics for higher perforated inserts along with decreasing the friction factor. Furthermore, a lower range of Reynolds number gives a higher value of TPF.
- The Nu varies between 60 to 210 with an increase in PI from 0% to 30% under the range of Reynolds number between 3000 and 21,000, whereas the friction factor varies between 0.05 and 0.51 with an increase in PI from 0% to 30% under the range of Reynolds number between 3000 and 21,000.

- The maximum TPF found is 3.25 at perforation index = 30% and *Re* = 3000. The enhancement in heat transmission is found 5.84 times better compared to the simple plain tube.
- It has also been observed from the study that this type of insert may be useful for industrial application because it significantly augments heat transmission by more than of five times and the thermal performance is enhanced by more than three times. The present study showed better heat transmission characteristics as compared to other research in the same range of Reynolds number.

**Author Contributions:** L.P.: Conceptualization, L.P.; methodology, L.P.; software analysis, L.P.; validation, L.P.; investigation, L.P.; writing—original draft preparation, L.P.; writing—review and editing, S.S.; visualization, S.S.; supervision, S.S. Both authors have read and agreed to the published version of the manuscript.

**Funding:** This research received no external funding.

**Conflicts of Interest:** The authors declare no conflict of interest.

## Nomenclature

| | |
|---|---|
| $A$ | Area of circular pipe ($m^2$) |
| $C_p$ | Specific heat of air ($Jkg^{-1}\,K^{-1}$) |
| $D$ | Test segment diameter ($m$) |
| $f$ | Friction factor for rough tube |
| $fs$ | Friction factor for simple plain tube |
| $h$ | Average convective heat transfer coefficient ($W\,m^{-2}\,K^{-1}$) |
| $k$ | Thermal conductivity of air ($Wm^{-1}\,k^{-1}$) |
| $L$ | Test section length ($m$) |
| $m$ | Air flow rate ($kgs^{-1}$) |
| $Nu$ | Nusselt number |
| $Nu_s$ | Nusselt number for simple plain tube |
| $P_A$ | Perforated area of insert ($m^2$) |
| $Pr$ | Prandtl number |
| $Q$ | Heat flux ($W$) |
| $Q_{air}$ | Heat carried by air ($W$) |
| $Q_{conv}$ | Heat transfer by convection ($W$) |
| $Re$ | Reynolds number |
| $T_A$ | Total area of Y-shaped insert ($m^2$) |
| $T_{fm}$ | Fluid mean temperature ($K$) |
| $T_i$ | Air temperature at inlet ($K$) |
| $T_o$ | Air temperature at outlet ($K$) |
| $T_{wm}$ | Wall mean temperature ($K$) |
| $v$ | Velocity of air ($ms^{-1}$) |
| $\Delta P$ | Pressure difference (between inlet and outlet of the test segment) ($Pa$) |

### Abbreviations

| | |
|---|---|
| TPF | Thermal performance factor |
| PI | Perforation index ($P_A/T_A$) |

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
