# Peer review of "Numerical Analysis for Heat Transfer Augmentation in a Circular Tube Heat Exchanger Using a Triangular Perforated Y-Shaped Insert"

_fluids, doi:10.3390/fluids6070247_

Round 1
Reviewer 1 Report
- Mathematical formulas are questionable in this form and there are discrepancies between text and some of equations.
- Authors is using turbulence model k- e, which is for the time-averaged Navier-Stokes equations (Reynolds Averaged Navier-Stokes) while momentum and energy equations are not time-averaged.
- Authors used k- e turbulence model, but they did not mention which one, as there are several k-e models, like for instance for low and high Reynolds number.
- Authors assumed that Reynolds number is changing from 3000 but did not proved that k-e model is suitable for such low Reynolds number.
- Description of geometry is not fully clear. It looks like wings are joining together in symmetry axis? More details or even picture would help.
- Mathematical definition of perforation index, which is 0%, 10%, 20% and 30%, should be added.
- The “Simple plain tube” should be defined.
- Authors did not make comparison of measurements with experiments even for smooth pipe. Such comparisons are important to verify, even partly, if the mathematical model is suitable.
- Authors did not show velocity field and pressure losses (dp/dx) in the circular pipe, which is important in analysing heat exchange.
Detailed amendments:
Abstract:
- Authors introduced abbreviation PI without definition.
- Authors used other abbreviations too without defining them. For instance, HE in Chapter 1.
- Authors used words: “a simple plain tube”, without defining what they really mean, as each tube possess own roughness. Is it smooth tube or ….? Clarification is required.
Chapter 2:
- Authors wrote that heat flux “Q” is 1000 W/m2. Authors should know that heat flux is in Watt, while density of heat flux “q” is in W/m2 or in the case of circular tube in W/m. Therefore, clarification is required.
Chapter 3:
- Equation, especially (3), (4) and (7), must be retyped in mathematical format using nominator and denominator, as in these forms they are questionable.
- Equation (6). It is not clear what temperatures Authors mean. Is it bulk temperature, i.e. averaged across the tube? Is it temperature between two wings or on a wing, etc?? This should be clarified also in Nomenclature.
Chapter 4.1:
- Before formulation of mathematical model, a Physical model should be stated first. All-important assumptions should be noted, like for instance: is it steady or unsteady flow, what coordinate system was used: cylindrical, Cartesian or other, which parameters depends on temperature, etc.
- Equations are not numbered.
- Continuity equation looks wrong, as there are not derivatives of velocity which is affected by wings and triangles. Only density is changing with position (x, y, z) which is in contradiction to this what Authors wrote in first sentence in the chapter: “… fluid flow is assumed to be atconstant fluid flow properties”.
Chapter 4.3:
- Grid independent tests are described too shortly. Authors talk about 884704 elements, and next show “Grid number” = 15,24,877. This Chapter is not clear and requires more efford from Authors.
- Authors should compare results of computations of Nusselt number with empirical formula at least for smooth circular tube.
Chapter 5:
Describing Fig. 3 Authors wrote: “For perforation index 30% case the enhancement in Nusselt number is 5.84 times over a simple plain tube”. From Fig 3 it looks like for PI=30% and Re ab. 21000 the Nu is about 220, which means it is ab. 70% higher comparing to “simple plain tube”.
Table 1, there is: “Pressure (atm)”.
Comment: This is not clear which pressure is it. Is it outlet pressure from circular tube or is atmospheric pressure? Clarification is required.
Author Response
Kindly see the attachment.

Reviewer 2 Report
Comments to the Authors
Title: Numerical analysis for heat transfer augmentation in circular tube heat exchanger using Triangular Perfo-rated Y-Shape Insert
The authors investigate numerically the heat transmission phenomenon in a tube heat exchanger having Y-shape insert with triangular perforation. The innovation of the present work is presented in a well-documented article but the comparison between experimental and numerical results and the validation of the proposed mathematical model is missing. Also, the presentation of conclusions is inadequate.
Following are the remarks. In view of the remarks, I recommend for major revision of the manuscript.
- The turbulence level at the entrance of the tube should be mentioned unless the flow is laminar.
- A documented qualitative or quantitative comparison between experimental and numerical results is missing. The validation of proposed mathematical model is necessary with corresponding experimental and numerical data because it determines the accuracy of computed measurements.
- Poorly written conclusions that do not allow the necessity and innovation of the present work to emerge and be adequately presented to stimulate further research.
Author Response
Kindly see the attachment.

Round 2
Reviewer 1 Report
I looked the corrected paper and generally the paper looks better then before. Therefore, the paper can be accepted.Reviewer 2 Report
The authors have carried out revisions. The manuscript is now acceptable for publication.